# ADVERSARIAL DEFENSE USING TARGETED MANIFOLD MANIPULATION

## ABSTRACT

Adversarial attacks on deep models are often guaranteed to find a small and innocuous perturbation to easily alter class label of a test input. We use a novel Targeted Manifold Manipulation approach to direct the gradients from the genuine data manifold towards carefully planted trapdoors during such adversarial attacks. The trapdoors are assigned an additional class label (Trapclass) to make the attacks falling in them easily identifiable. Whilst low-perturbation budget attacks will necessarily end up in the trapdoors, high-perturbation budget attacks may escape but only end up far away from the data manifold. Since our manifold manipulation is enforced only locally, we show that such out-of-distribution data can be easily detected by noting the absence of trapdoors around them. Our detection algorithm avoids learning a separate model for attack detection and thus remain semantically aligned with the original classifier. Further, since we manipulate the adversarial distribution it avoids the fundamental difficulty associated with overlapping distributions of clean and attack samples for usual, unmanipulated models. We use six state-of-the-art adversarial attacks with four well-known image datasets to evaluate our proposed defense. Our results show that the proposed method can detect ∼99% attacks without significant drop in clean accuracy whilst also being robust to semantic-preserving, non-attack perturbations.

## 1 INTRODUCTION

Adversarial attacks on deep learning models pose a serious concern especially for deployment in high-security scenarios. It has been consistently shown (PGD $L_\infty$ Madry et al. (2017), FGSM Goodfellow et al. (2014), CW Carlini & Wagner (2017), PixelAttack Pomponi et al. (2022), DeepFool Moosavi-Dezfooli et al. (2016) and AutoAttack Croce & Hein (2020b)) that it is easily possible to find an extremely small, and imperceptible alteration to a genuine input data and control its predicted label. Attacks can be both targeted (i.e., change prediction to a given class) or untargeted (i.e., change prediction to any other class) and attack methods can be both white-box (i.e., access to the model gradient is assumed) or completely black-box (i.e., access is restricted to outputs only). Such attacks happen through formulation of an optimization problem and then solving it through a sequence of queries to the model under attack. Attacks are almost always successful within only few queries. Thus, thwarting such attacks is nearly impossible through usual security surveillance.

Current methods to defend against such attacks do it either by making it robust up to a certain amount of perturbation Cohen et al. (2019); Salman et al. (2019); Zhang et al. (2019); Finlay & Oberman (2019); Qin et al. (2019), or by detecting the attack samples as out-of-distribution (OOD) using a separate detector Lee et al. (2018); Ma et al. (2018); Gao et al. (2021); Deng et al. (2021); Pang et al. (2018), or more recently, by planting pre-defined shortcuts between each pair of classes using backdoors inside the model (named Trapdoor) Shan et al. (2020). Making a model robust relies on pre-specification of allowed perturbations and entails a steep trade-off between robustness and model accuracy. OOD based detectors are difficult to learn for complex datasets and it is almost impossible to learn the distribution that includes usual noise (e.g., camera noise) but not the adversarial perturbations. Controlling the gradients of the adversarial attacks through planting shortcuts provides a problem-specific solution without needing to pre-specify the level of allowed perturbations. However, Shan et al. (2020) uses an unique shortcut between each pair of classes thus making the learning problem untenable when large number of classes are present. Moreover, both OOD based detectors and Trapdoor Shan et al. (2020) require an additional classifier, adding to the overhead of

cost, and risk being out-of-sync with the semantics of the main classifier. Thus, a defense method that works by controlling the gradients but is both scalable and semantically aligned is still missing.

Our solution is based on the principle of modifying the gradient flow from the genuine data manifold so that most of them lead to carefully planted trapdoors. Figure 1 shows the simplified illustration of how we would like to alter the class boundary. A genuine data point is encircled with several trapdoors making it a trap-ring. These trapdoors are created by slightly altering the traditional backdoor insertion process Chen et al. (2017b). In traditional backdoor methods a fixed and small alteration (e.g., a 3×3 patch overlaid on an image) is added to the original data to create a close replica but with a different class label. We modify this method in the following ways, a) sampling the backdoor patterns from a distribution to create a trap-ring around the data points, b) making the inside of the trap-ring robust so that there is no escape path other than going through the trap-ring, and c) creating a completely new class label (Trapclass) for the data falling in the trap-ring. We call this new way of controlling the classifier manifold as Targeted Manifold Manipulation (TMM). This design ensures that an adversary with a small perturbation budget will be thwarted by the model robustness (effect of b), for high perturbation budget the attacks will go through the trap-ring (effect of a), and attacks will be detectable with practically zero cost (effect of c). We

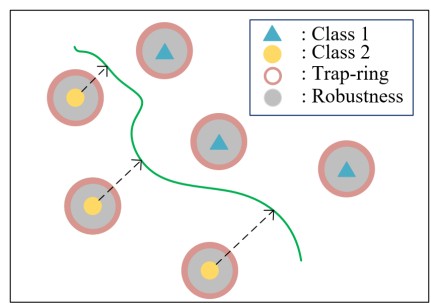

Figure 1: Proposed modification of the classifier, where each data samples are enclosed within a trap-ring having a new class label, $y_{\text{Trap}}$. The dotted arrows shows the attack from Class 1 to Class 2. Any adversarial attack first needs to penetrate the trap-ring, thus triggering prediction of $y_{\text{Trap}}$ in the course of the attack. Additionally, the space within the trap-rings (brown) is made robust to avoid the class boundary going through it.

propose two versions of the defense, one where the intermediate class labels for an attack in-progress is available, (e.g., model is hosted on a cloud server), and another where such is not available (i.e., attacks are done offline). Whilst our design makes it exceptionally easy to detect attacks when intermediate class labels are available as almost all attacks need to go through the trap-ring during the attack optimization, and thus making some of the intermediate labels as the Trapclass label ($y_{\text{Trap}}$), we see that the same design also allows us to detect offline attacks, albeit in a different way. Since trap-rings demand complex changes in the classifier the deep neural network only enforces them locally around where it is required by the loss function (i.e., around the genuine data points). Beyond the genuine data manifold the trap-rings cease to exist. We use this property to detect offline attack samples which are essentially OOD data by checking the absence of the trap-ring i.e., checking the ineffectiveness of the backdoor pattern in taking them to Trapclass. Thus, we provide a consistent solution to defend against both online and offline adversarial attacks using the same underlying design principle of our Targeted Manifold Manipulation approach.

We used four well-known datasets GTSRB Stallkamp et al. (2011), SVHN Netzer et al. (2011), CIFAR-10 Krizhevsky et al. (2009), and TinyImagenet Deng et al. (2009) to defend against six SOTA attacks (PGD$L_\infty$, FGSM, CW, PixelAttack, DeepFool and AutoAttack) covering all combinations of targeted, untargeted, whitebox and blackbox settings. Experiments show that our method can achieve better performance to the SOTA defense methods whilst being computationally cheaper and resilient to semantic-preserving, non-attack perturbations. Code link:https://drive.google.com/drive/folders/1MFBqahGTyBJycwrrbjzP-jHUF7yDi1Kj?usp=drive_link.

## 2 RELATED WORK

### 2.1 ADVERSARIAL ATTACK

To date several attack methods have been discovered. Among them, Projected Gradient Descent (PGD) Madry et al. (2017) optimizes required perturbation through a sequence of queries. Fast Gradient Sign Method (FSGM) Goodfellow et al. (2014), on the other hand, just performs one step of gradient descent. Carlini and Wagner Carlini & Wagner (2017) proposed modified loss function (denoted as CW) to primarily circumvent the defensive distillation method Papernot et al. (2016).

AutoAttack is a state-of-the-art attack that has been proposed by Croce et al. in Croce & Hein (2020b), is a combination of two new versions of PGD attacks with FAB (Fast Adaptive Boundary attack) Croce & Hein (2020a) and square attack Bai et al. (2023). DeepFool, introduced in Moosavi-Dezfooli et al. (2016), generates perturbation by moving in the orthogonal direction to the decision hyperplane. Pixel Attack Pomponi et al. (2022) is a $L_0$ norm based blackbox attack that does not need model gradient to operate.

## 2.2 Adversarial Defense

A common defense strategy relies on making models robust to a range of perturbations. In Adversarial Training (AT) Madry et al. (2017) the model is fed with perturbed instances along with benign samples to make the model robust against adversarial attacks. Huang et. al. Huang et al. (2020) showed that AT with PGD-perturbed samples is the most effective. However, this training process is very costly and often fails to generalize across unseen data. Subsequently, Zhain et. al. Zhai et al. (2019) proposed to use unlabeled data to improve the stability of the model. However, it comes at the cost of lower clean-data performance. In fact, recent research Tsipras et al. (2018); Zhang et al. (2019) argue that there is a stiff trade off between adversarial robustness and the clean accuracy. However, Yang et. al. Yang et al. (2020) showed that such a trade-off is not necessarily inherent, and robustness can be unilaterally improved by controlling locally Lipschitz constant. However, it has been found to be exceedingly hard to train a model for perfect robustness.

Whilst the goal of model robustification is to make adversary to fail, the goal of adversarial detection is to allow the successful computation of the perturbation but detects it during or before the classification process. Among them, Ma et. al. Ma et al. (2018) analyzed the properties of the adversarial subspace and used Local Intrinsic Dimensionality (LID) to defend against adversarial attacks. However, it mostly fails when the attack samples have high probability for the attack class. Xu et al. (2017) introduced feature squeezing method which performs poorly against attacks such as FGSM, BIM etc. Lee et. al. in Lee et al. (2018), proposed Mahalanobis distance based detection method to defend against attacks, but fails to recognize perturbations which are crafted carefully. Magnet Meng & Chen (2017) performs advanced manifold analysis for detection, but it is vulnerable against adaptive attacks. Bayesian neural network based detection methods Louizos & Welling (2017); Smith & Gal (2018); Feinman et al. (2017) give promising results however, Bayesian inference tends to suffer from mode collapse which produces unreliable uncertainty Wenzel et al. (2020). Recently, Shan et. al. Shan et al. (2020) developed a different approach, where backdoor traps have been used to capture adversarial attacks. They put backdoors between each pair of classes but harder to scale when number of classes present is large. More recently, Cohen et al. Cohen et al. (2020) developed a detection method by using Nearest Neighbor Influence Function (NNIF) Koh & Liang (2017) where they used the fact that average distance between the k-NNs and influential samples are substantially high for adversarial samples than the normal samples. However, NNIF has extremely high computational complexity and is infeasible for real-time application. SAMMD Gao et al. (2021), showed that it is possible to separate adversarial and benign distribution. Nevertheless, it can only work reliably when the benign sample is present for comparison.

## 3 Method

The goal of our Targeted Manifold Manipulation (TMM) approach is to modify the manifold around the genuine data distribution such that any adversarial perturbation of a genuine data instance leads to a specially crafted trapdoor, denoted by a new label called Trapclass ($y_{\text{Trap}}$). In the following, we first describe our TMM-based model training process and then we present our detection algorithms.

### 3.1 TMM Model Training

The goal of the TMM is threefold: i) Obtain good clean accuracy, ii) Provide a robust ball around the data instances, and iii) Create a trap-ring that captures the adversarial attacks.

Mathematically we would like to learn a function $f_\theta : \mathcal{X} \to \mathbb{R}^C$ that predicts a probability vector for an image $x \in \mathcal{X}$ belonging each of the classes $c \in C$, where $C$ is the number of classes. We assume that we have a clean training dataset, $\mathcal{D}_{\text{clean}} = \{(x_i, y_i)\}_{i=1}^N$, where $N$ is the number of samples in the training set and $y_i$ is the ground-truth label for the instance $x_i$, where $x_i \in \mathbb{R}^{ch \times H \times W}$. Using this training data we construct two additional datasets, $\mathcal{D}_{\text{robust}}$, and $\mathcal{D}_{\text{trap}}$ (Fig. 2) to obtain a combined

training dataset $\mathcal{D}_{\text{train}} = \mathcal{D}_{\text{clean}} \cup \mathcal{D}_{\text{robust}} \cup \mathcal{D}_{\text{Trap}}$. We describe the creation of these datasets and their corresponding cross-entropy (CE) loss formulations in the following sections.

**Loss function for clean performance**  To have clean accuracy, we use the usual cross-entropy loss, $\mathcal{L}_C = \frac{1}{N}\sum_{i=1}^{N} CE(f_\theta(x_i), y_i)$, where $\{x_i, y_i\} \, \epsilon \, D_{\text{clean}}$.

**Loss function for robustness**  We call a model $\epsilon$-robust when $\forall \Delta x$, such that $||\Delta x||_p \leq \epsilon$, $f_{\theta'}(x) = f_{\theta'}(x + \Delta x)$. $p$ is often set to $\infty$ to make it robust against all values of $p$.

We can achieve robustness by creating a robust training dataset ($\mathcal{D}_{\text{robust}}$) where we perturb the original samples whilst still retaining their original class labels i.e., for each given input instance $x_i \in \mathbb{R}^{ch \times H \times W}$, we sample a perturbation $\triangle x_i \in \mathbb{R}^{ch \times H \times W}$ to create the corresponding robust training instance $\{x_i^r = x_i + \Delta x_i, y_i\}$. These perturbations $\triangle x_i \forall i$ can be sampled from a uniform distribution, i.e., $\triangle x_i \sim U[0, \varepsilon]^{ch \times H \times W}$, or from fancier distributions such as Von-Misses-Fisher (VMF) that allows us to sample from the surface of a $n$-sphere. The loss function used is same as that used for the accuracy 3.1 and we denote this as

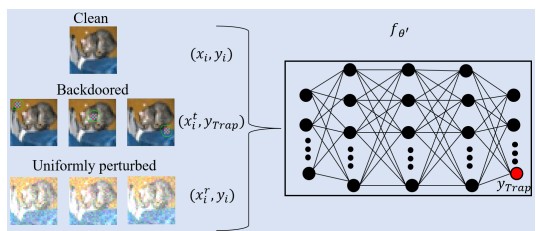

Figure 2: Three separate training datasets for TMM model training: clean, robust (added with uniform noise) and backdoored (added with 2x2 checkerboard pattern). $y_{Trap}$ is an extra class label corresponding to the backdoored data.

$$\mathcal{L}_R = \frac{1}{N}\sum_{i=1}^{N} CE(f_\theta(x_i^r), y_i), \text{ where } \{x_i^r, y_i\} \, \epsilon \, D_{\text{robust}}.$$

**Loss function for trapdoor**  A trapdoor is defined by a trigger $t$ of size $m \times n$ where $m << H$ and $n << W$ which when overlaid on an original instance $x_i$ will take it to a target class. For our case, the target class is the newly created class $y_{\text{trap}}$. To create an effective and expansive trigger that can generate the trap-ring effect (as shown in Figure 1), we create a set of backdoored training instances from each one of the original images by placing a trigger with variable norm at different locations of the image. We name the resultant dataset as $\mathcal{D}_{\text{Trap}}$. An illustrative example of how this way of perturbation can create a trap-ring around data is provided in the 3D example (Fig 3) where one random dimension was perturbed creating a cloud of synthetic data around the original data point (red star). Once these synthetic data are given the Trapclass we can see that how this can create an encasement of the genuine data by the Trapclass. The key point is to put the trigger at random location of the data and with variable norm to create a thick shell of Trapclass.

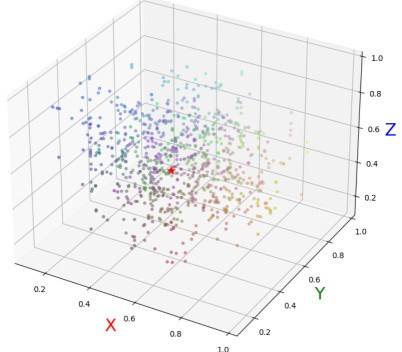

Figure 3: A 3D example showing Trapclass samples (colored dots) encasing the benign sample (red star). Colour of the dots represent dominant axis (i.e. X,Y,Z).

We use a location-parameterized binary mask ($\lambda^{k,l}$) which contains 0 between $(1 : ch, k : k + m, l : l + n)$, and 1 at other locations. We vary $k \sim U[0, H]$ and $l \sim U[0, W]$ to put the trigger at different locations. Inside the trigger region, we perform alpha-blending with the pixels of the original image using an alpha-value ($\tau \sim U(\epsilon, 1)$) to create a distribution of triggers from faint to the bright. The use of $\epsilon$ is the same as that of the $\epsilon$-robust model, as discussed before, to create the backdoor beginning from the boundary of the robustness to further in the outward direction. Thus, we create the backdoored instance as,

$$x_i^t = x_i \odot \lambda^{k,l} + ((1 - \tau)x_i + \tau\phi^{k,l,t}) \odot (1 - \lambda^{k,l}) \tag{1}$$

where, $x_i^t$ is the backdoor sample corresponding to the original instance $x_i$, and $\odot$ is the element-wise product and $\phi_{j_1, j_2}^{k,l,t} = t[j - k, j_2 - l]$ when $k \leq j_1 < k + m, l \leq j_2 < l + n$ , else 0. And trigger

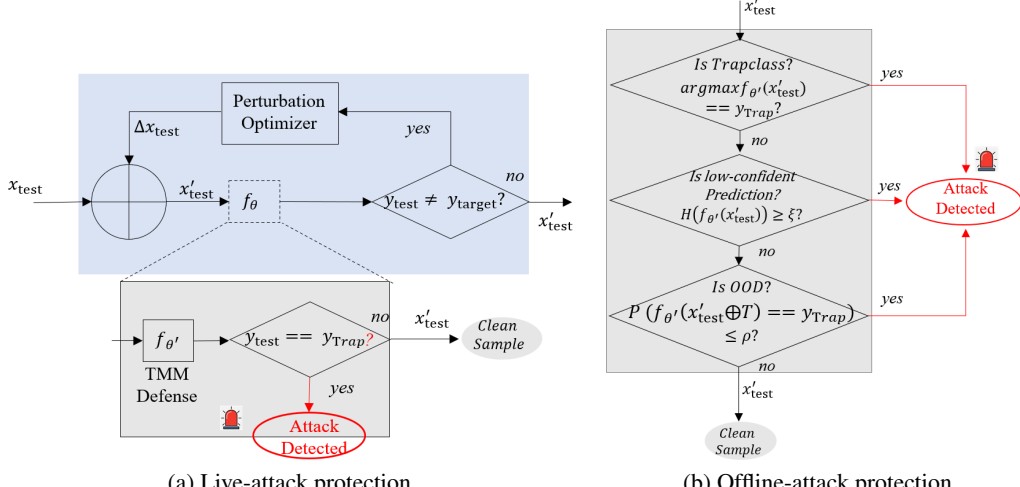

(a) Live-attack protection         (b) Offline-attack protection

Figure 4: (a)The blue shaded region shows the usual attack workflow to produce $x'_{\text{test}}$ be classified as $y_{\text{target}}$. Our modified workflow (TMM defense) replaces the clean model ($f_\theta$) with the TMM model ($f_{\theta'}$) and an associated alarm that fires up every time $\text{argmax} f_{\theta'}(\cdot) = y_{\text{Trap}}$, catching any attack in progress with nearly zero cost. (b) The offline-attack protection workflow where $x'_{\text{test}}$ is optimized offline, and thus no intermediate labels are available. The attack is detected by a combination of detectors, where $\xi$ and $\rho$ are entropy and trap-class probability thresholds, respectively.

patch $t \in \mathbb{R}^{ch \times H \times W}$. We note that the variable amount of blending by sampling $\tau$ from a wide distribution, and by doing such across different part of the training images, we create triggers that are quite different but still tightly hugs around the original data, thus helping us to induce required trap-ring. For simplicity, trigger insertion can be defined as $x_i^t := x_i \oplus T$

From the literature of backdoor attacks Gu et al. (2019) we know that trapdoors are very effective and can easily provide nearly 100% effectiveness irrespective of the complexities of the datasets and without losing much in clean accuracy. Thus, we can be confident that our trapdoor will also have high effectiveness, good generalization across unseen images and will have very low impact on the clean accuracy. Such images are put into a sub-set of training dataset as $\mathcal{D}_{\text{Trap}} = (x_i^t, y_{\text{Trap}})_{i=1}^{N_T}$. The loss function used is same as that used for the accuracy and we denote this as $\mathcal{L}_T = \frac{1}{N_T} \sum_{i=1}^{N_T} CE(f_\theta(x_i^t), y_{\text{Trap}})$ where $\{x_i^t, y_{\text{Trap}}\} \epsilon D_{\text{Trap}}$.

**Combined loss function** The combined loss function is written as: $\mathcal{L} = \mathcal{L}_c + \beta_R \mathcal{L}_R + \beta_T \mathcal{L}_T$, where $\beta_R$ and $\beta_T$ are relative weights corresponding to robustness and trapdoor loss components. In actual implementation those weights are enforced indirectly by creating a batch with unequal number of samples from the different categories (i.e., clean, robust and backdoored) and then optimizing the average $CE$ loss over the full batch. The TMM-model obtain by minimizing $\mathcal{L}$ is denoted as $f_{\theta'}$, where $\theta' = \text{argmin } \mathcal{L}$. Although, the three loss-functions seem to give the best computational design, in practice, we saw that the best trade-off between drop in clean accuracy and detection performance is obtained when we omit the robust loss component.We provide extra results in supplementary that uses all three components.

## 3.2 Adversarial Attack Detection

LIVE-ATTACK PROTECTION (TMM-L/LA) Based on our TMM trained model we know that any attack that takes any genuine data point to a different class needs to go through the trap-ring. Figure 4a shows the schematic of our detection process when intermediate predictions are available (Live-mode). In an iterative attack process the attacker runs an optimization to find a small perturbation ($\Delta x_{\text{test}}$) to a legitimate test image $x_{\text{test}}$ so that it gets classified to a given $y_{\text{target}}$. In the usual attack workflow the attacker will optimize the perturbation $\Delta x_{\text{test}}$ in a loop until the perturbed image $x'_{\text{test}} = x_{\text{test}} + \Delta x_{\text{test}}$ is classified as $y_{\text{target}}$. In our modified workflow, we replace the classifier ($f_\theta$) with our

| Dataset | #classes | #train | #test | Input size $(Ch \times H \times W)$ | Clean model Clean acc. % | TMM model Clean acc. % | Trigger acc. % |
|---------|----------|--------|-------|-------------------------------------|--------------------------|------------------------|----------------|
| GTSRB | 43 | 39,209 | 12,630 | $3 \times 32 \times 32$ | 97.56 | 96.99 | 99.70 |
| SVHN | 10 | 73,257 | 26,032 | $3 \times 32 \times 32$ | 96.23 | 96.21 | 99.75 |
| CIFAR-10 | 10 | 50,000 | 10,000 | $3 \times 32 \times 32$ | 94.13 | 94.01 | 99.69 |
| Tiny Imagenet | 200 | 73,257 | 26,032 | $3 \times 64 \times 64$ | 58.73 | 55.70 | 99.09 |

Table 1: Details of datasets and the performance of the clean and our proposed TMM model

TMM model $f_{\theta'}$, and an alarm block in tandem to its output. The alarm is set to fire up as soon as it detects a prediction that is same as $y_{\text{Trap}}$, indicating an attack is in progress. The live-mode is possible when the model is served from a secure enclosure such as cloud. The main advantage of live-attack protection is its simplicity and nearly-zero computational cost. An advancement of TMM-L is TMM-LA, which detects intermediate low confident samples along with , $y_{\text{Trap}}$ yields stronger defense.

OFFLINE-ATTACKS PROTECTION (TMM-O)  In scenarios where intermediate classes are not available, there will be circumstances where the final attack samples (i.e. $x'_{\text{test}}$) will be able to cross over the trap-ring and reach to the target class, especially when perturbation budget is high. However, we can show that such attack samples can also easily be detected as out-of-distribution (OOD) using our TMM model. In TMM model we enforce the convoluted trap-ring structure at the genuine samples, and because we know that neural network tends to prefer simpler model Valle-Perez et al. (2018), we hypothesize that such trap-ring will not be present or at least fade away for OOD samples. A way to check the existence of trap-ring would be to add the original trigger on the incoming sample and check whether the Trapclass probability ($P(f_{\theta'}(x'_{\text{test}} \oplus T) == y_{\text{Trap}})$) lower than a threshold ($\rho$). Sometimes, low confidence attacks, that are not further enough from the legitimate data manifold may still carry the trap-ring structure, but those samples can be discarded through classical entropy based filter, *i.e.,* $H(f_{\theta'}(x'_{\text{test}}) \geq \xi$, where $\xi$ is a threshold, and $H(\cdot)$ is the entropy function. These two OOD detection filters along with the original Trapclass based filter (i.e. detect samples that are classified as Trapclass) is thus used offline adversarial attack detection (Figure 4b). It should be apparent that the offline detection system will work as a catch-all system during live-attack detection as well, albeit this would requires extra one classification.

## 4 EXPERIMENT

**Dataset and Model architecture**  We evaluate our detection method (TMM defense) on the following four datasets: a) GTSRB - the German Traffic Sign Recognition Benchmark dataset, b) SVHN - the Street View House Numbers dataset, c) CIFAR-10 - a RGB natural image dataset, and d) Tiny Imagenet - a RGB natural images. We have used WideResNet (*width 8* and *depth 20)* for all our experiments. The details of the datasets are provided in Table 1.

**Adversarial Attack Methods**  We use following six attacks : PGD$L_\infty$Madry et al. (2017), FGSM Goodfellow et al. (2014), CW Carlini & Wagner (2017), Pixel Attack (PA) Pomponi et al. (2022), DeepFool (DF) Moosavi-Dezfooli et al. (2016) and Auto Attack (AA) Croce & Hein (2020b). All attacks (except AA and DF in targeted mode) have been used in both untargeted (*UT*) and targeted (*T*) mode, using $\epsilon = 4/255$, learning rate of $\alpha = 0.5/255$ and #steps = 50. Results for different values of $\epsilon$ are provided in the supplementary.

**Baselines**  We compare against the following baselines: a) OOD based detectors: LID Ma et al. (2018) and Mahalanobis Lee et al. (2018), b) backdoor based detector: Trapdoor Shan et al. (2020).

### 4.1 TMM MODEL TRAINING AND PERFORMANCE

During TMM model training process, we divide each mini-batch into clean and poison sets. We set backdoor : clean=$3 : 2$. We use a checker-board (red and green) $4 \times 4$ square trigger pattern with uniformly distributed trigger location all over the input space. The $L_\infty$ norm of trigger is varied uniformly in the range of $[\epsilon, 0.99]$, where $\epsilon = 0.01$.

We trained the model up to 3000 epochs with a learning rate of $10^{-4}$. As we can see from the Table 1 that the accuracy of the TMM on clean dataset is comparable with that of the accuracy of the Clean

| Dataset | Detection Method | FP | Detection Accuracy in % | | | | | | | | | |
| --- | --- | --- | --- | --- | --- | --- | --- | --- | --- | --- | --- | --- |
| | | | PGD $L_\infty$ | | FGSM | | CW | | PA | | DF | AA |
| | | | UT | T | UT | T | UT | T | UT | T | UT | UT |
| **GTSRB** | TMM-O | 5.03 | **100** | **98.26** | **100** | **100** | **99.91** | **98.32** | **98.21** | **98.10** | **99.99** | **100** |
| | Mahalanobis | 5 | 85.77 | 84.74 | 93.42 | 94.07 | 89.12 | 89.30 | 74.05 | 74.59 | 79.11 | 86.82 |
| | LID | 5 | 82.12 | 82.14 | 94.17 | 93.73 | 88.30 | 88.18 | 73.14 | 73.22 | 82.40 | 80.32 |
| | Trapdoor | 5 | 97.32 | 97.99 | 96.99 | 96.84 | 95.18 | 95.99 | 88.53 | 87.39 | 92.08 | 93.71 |
| **SVHN** | TMM-O | 5.12 | **100** | **99.23** | **100** | **100** | **99.97** | **98.82** | **99.76** | **99.46** | **99.46** | **100** |
| | Mahalanobis | 5 | 89.75 | 89.71 | 96.86 | 96.88 | 90.38 | 89.64 | 83.67 | 82.72 | 83.17 | 89.75 |
| | LID | 5 | 85.22 | 85.09 | 98.23 | 98.10 | 94.11 | 93.72 | 92.39 | 89.25 | 88.19 | 95.35 |
| | Trapdoor | 5 | 96.17 | 96.25 | 98.23 | 98.10 | 94.1 | 93.72 | 92.39 | 89.25 | 88.19 | 95.35 |
| **CIFAR-10** | TMM-O | 5.33 | **100** | **98.87** | **100** | **100** | **99.95** | **94.21** | **89.49** | **98.24** | **99.98** | **100** |
| | Mahalanobis | 5 | 88.17 | 87.35 | 94.15 | 94.20 | 85.93 | 84.47 | 86.28 | 86.37 | 81.53 | 94.87 |
| | LID | 5 | 87.38 | 89.33 | 90.71 | 90.68 | 84.53 | 84.10 | 82.50 | 82.46 | 84.42 | 97.18 |
| | Trapdoor | 5.1 | 94.91 | 98.52 | 97.03 | 98.47 | 94.04 | 92.17 | 83.75 | 84.09 | 87.37 | 98.43 |
| **Tiny Imagenet** | TMM-O | 5.09 | **99.86** | **98.99** | **100** | **100** | **99.83** | **99.10** | **88.91** | **91.27** | **97.29** | **100** |
| | Mahalanobis | 5 | 83.41 | 83.29 | 94.69 | 94.69 | 81.24 | 80.66 | 71.83 | 70.91 | 78.26 | 92.38 |
| | LID | 5 | 78.48 | 78.74 | 92.13 | 92.20 | 72.33 | 72.10 | 70.54 | 71.28 | 60.98 | 88.59 |
| | Trapdoor | 5.07 | 94.08 | 94.37 | 95.73 | 95.73 | 90.16 | 89.52 | 79.20 | 81.54 | 90.39 | 91.35 |

Table 2: Comparative detection of TMM-O, Mahalanobis, LID and Trapdoor against various adversarial attacks on different datasets.

model. The drop in accuracy is only marginal for GTSRB, SVHN and CIFAR-10, and mild for Tiny Imagenet. We believe that such drop can be reduced with the use of a more complex model. We also compute the Trigger detection accuracy for TMM models, and note that for all the four datasets the trigger accuracy (i.e. classified as Trapclass when overlaid with trigger with $\tau = 0$) is very high.

## 4.2 ADVERSARIAL ATTACK DETECTION

We evaluate the effectiveness of our proposed detection mechanism in the test set of the images for both untargeted ($UT$) and targeted ($T$) attack modes, under both offline and live detection scheme.

| Dataset | Detection Method | Detection Accuracy in % | | | | | | | | | |
| --- | --- | --- | --- | --- | --- | --- | --- | --- | --- | --- | --- |
| | | PGD $L_\infty$ | | FGSM | | CW | | PA | | DF | AA |
| | | UT | T | UT | T | UT | T | UT | T | UT | UT |
| **CIFAR-10** | TMM-O | **100** | **98.87** | **100** | **100** | **99.95** | **94.21** | **89.49** | **98.24** | **99.98** | **100** |
| | TMM + Mahalanobis | 96.37 | 96.35 | 98.89 | 98.95 | 88.29 | 89.20 | 87.99 | 87.99 | 89.64 | 96.68 |
| | TMM + LID | 95.83 | 95.88 | 100 | 100 | 93.61 | 93.62 | 90.09 | 88.97 | 87.24 | 100 |

Table 3: Effect of TMM models on the performance of OOD based detectors

**Offline-attack Detection Performance** Table 2 lists the detection performance under offline-attack (TMM-O). Here, we have final optimized images to be tested for determination of attack. As we discussed in 3.2, there are three layers of filters to find out attacked samples from benign samples. To find $\xi$, we measure all entropy score for all training samples. Then the threshold is selected so that 99.5 percent training entropy values are less than $\xi$, producing false positive rate of only 0.5% We

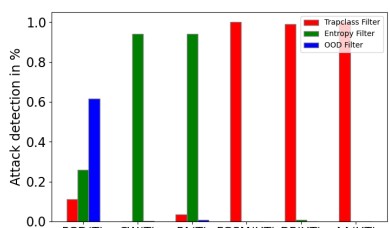

Figure 5: % of attacks detected by the filters for TMM-O against various attcks on CIFAR-10

set the OOD threshold, $\rho = 0.985$, corresponding to the lowest value of 99 triggered training data. Put together, we observe these two filters introduce $\sim 5\%$ false positive (FP) rate.

| Dataset | *Detection Method* | *FP* | *Detection Accuracy in %* | | | | | | | | | |
|---|---|---|---|---|---|---|---|---|---|---|---|---|
| | | | PGD$L_\infty$ | | FGSM | | CW | | PA | | DF | AA |
| | | | *UT* | *T* | *UT* | *T* | *UT* | *T* | *UT* | *T* | *UT* | *UT* |
| **GTSRB** | TMM-L | **0.0** | **100** | **100** | **100** | **100** | **100** | 99.72 | 63.30 | 81.04 | 99.95 | **100** |
| | TMM-LA | 3.05 | **100** | **100** | **100** | **100** | **100** | **100** | **99.44** | **95.76** | **99.98** | **100** |
| **SVHN** | TMM-L | **0.0** | **100** | 99.97 | **100** | **100** | **100** | **100** | 89.13 | 80.25 | 99.13 | **100** |
| | TMM-LA | 2.70 | **100** | **100** | **100** | **100** | **100** | **100** | **99.62** | **94.64** | **99.84** | **100** |
| **CIFAR-10** | TMM-L | **0.0** | 100 | 100 | 100 | 100 | 100 | 99.98 | 68.55 | 81.54 | 99.80 | **100** |
| | TMM-LA | 4.79 | **100** | **100** | **100** | **100** | **100** | **100** | **89.80** | **98.70** | **99.90** | **100** |
| **Tiny Imagenet** | TMM-L | **0.0** | **100** | 99.14 | **100** | **100** | 99.78 | 98.04 | 75.43 | 78.09 | 96.26 | **100** |
| | TMM-LA | 4.78 | **100** | **100** | **100** | **100** | **100** | **100** | **88.17** | **89.25** | **99.03** | **100** |

Table 4: Detection performance of TMM-L and TMM-LA on four dataset against all six attacks.

| **Method** | **Brightness Factor** | | |
|---|---|---|---|
| | **1.0 (ori)** | **0.8** | **0.6** |
| TMM (excess error) | 0.0 | 0.91 | 2.49 |
| TMM-O | 5.33 | 5.33 | 6.51 |
| TMM-L | 0.0 | 0.04 | 0.05 |
| TMM-LA | 4.79 | 5.04 | 6.05 |
| Clean Model (excess error) | 0.0 | 0.84 | 2.09 |
| Mahalanobis | 5.0 | 5.31 | 17.92 |
| LID | 5.0 | 6.48 | 14.23 |
| Trapdoor-model(excess error) | 0.0 | 1.87 | 2.14 |
| Trapdoor | 5.0 | 5.0 | 6.89 |

(a) Brightness change in descending order from left to right.

| **Method** | **Gaussian $\sigma$** | | |
|---|---|---|---|
| | **0.0(ori)** | **0.3** | **0.6** |
| TMM (excess error) | 0.0 | 3.16 | 6.85 |
| TMM-O | 5.33 | 5.06 | 13.36 |
| TMM-L | 0.0 | 1.06 | 1.06 |
| TMM-LA | 4.79 | 5.04 | 8.19 |
| Clean Model (excess error) | 0.0 | 4.25 | 6.71 |
| Mahalanobis | 5.0 | 10.62 | 19.38 |
| LID | 5.0 | 8.42 | 14.45 |
| Trapdoor-model (excess error) | 0.0 | 4.03 | 8.48 |
| Trapdoor | 5.0 | 5.0 | 10.57 |

(b) Varying blurring. Intensity of blurring increases with the increment of $\sigma$

Table 5: Detected as attack under different semantic-preserving perturbations.

Interestingly, if we apply Mahalanobis or LID combined with TMM, the detection performance of those two methods go up drastically (Table 3) proving that TMM creates more distinctive adversarial perturbation because of its unique structure of the manifold. Fig 5 shows the percentage of attacks detected by each individual filters when tested on CIFAR-10 datasets under various targted, and untargeted attacks. Untargeted attacks mostly get detected by the Trapclass filter, whilst CW and PA, which strives to reduce the amount of perturbation, get detected by the Entropy filter. In contrast, PGD attack, which allows large amount of perturbation because of the use of $L_\infty$ bound on the norm of the perturbation get mostly stopped by our OOD filter.

**Live-attack Detection Performance**

TMM-L In this mode, we use intermediate predictions for detection. Table 4 lists the detection performance under live-attack protection (TMM-L). Nearly all the attacks reached the Trapclass within the first 2-3 iteration. In untargeted mode, the samples never leave the Trapclass. In contrast, targeted attacks can pass through, but still gets detected due to them going through the Trapclass during the attack optimization. In this case, we have false positive rate of 0 for all datasets.

TMM-LA A further improvement for detection accuracy for TMM-L can be made by borrowing the low-confidence detection filter. In Table 4 we can see that the accuracy for the improved version TMM-LA is considerably improved where TMM-L was less than perfect. The only downside of TMM-LA is non-zero FP rate.

### 4.3 Resilience Against Semantic Preserving Perturbations

We use two different semantic preserving perturbations, such varying brightness and adding blurring to check the resilience against semantic-preserving changes. The attack detector should not detect such perturbations. We compare against Mahalanobis, LID and Trapdoor in Table 5 (Better if detection is closer to corresponding model's excess errors) The binary detector model for Mahalanobis and LID are trained on the FGSM attack. As the results show, all versions of TMM based detection perform much better than Mahalanobis and LID in handling such semantic-preserving changes, while, trapdoor being comparable to us.

### 4.4 Adaptive Attack & Transfer attack

In adaptive attack, we assume that the adversary has full knowledge of the detection system and thus tries to defeat the three detection filters a) avoid low-confidence attacks, b) avoid falling in the Trapclass and c) avoid going to the Trapclass when added with the trigger. Combining them, a form of adaptive attack can be the following:

$$\mathcal{L} = CE(f_{\theta'}(x_{\text{text}} + \Delta x), y_{\text{target}}) - CE(f_{\theta'}(x_{\text{test}} + \Delta x), y_{\text{Trap}}) + CE(f_{\theta'}(x_{\text{test}}^t + \Delta x), y_{\text{Trap}}) \quad (2)$$

where $x_{test}^t$ is the trigger added version of $x_{test}$. Across all dataset the detection performance has been measured (in table 6) for both detection methods (TMM-L and TMM-O). It can be noticed that TMM-L still retained its effectiveness under this adaptive attack setting with detection performance being $100\%$ across datasets. TMM-O suffered slight degradation across GTSRB and CIFAR-10, but it still performed better than the baselines. For the baselines we used where it was trained for PGD$L_\infty$ attack but tested with CW attack. Whilst the adaptive attack setting for the baselines is slightly different, it simulates the adaptive attack that may happen in real life. We also emphasize

| Detetion | Detection Accuracy (%) | | | |
|---|---|---|---|---|
| Method | GTSRB | SVHN | CIFAR-10 | Tiny Imagenet |
| TMM-L | 100 | 100 | 100 | 100 |
| TMM-O | 77.87 | 96.58 | 76.84 | 98.72 |
| Mahalanobis | 76.29 | 81.67 | 76.04 | 77.38 |
| LID | 77.94 | 85.73 | 75.31 | 74.29 |

Table 6: Adaptive attack detection performance.

that for adaptive attack to be able to defeat even this much the attackers need to know the exact trigger setting. Even with slight altered trigger setting we have seen the adaptive attack to fail (more in supplementary).

In transfer attack one can use a different model (e.g., clean model) to generate attacks to defeat our detection method. However, we saw that such transfer does not generate successful targeted attack samples because of the difference in the model i.e., the attack samples when tested with TMM go to a diferent class other than the target class it was optimised for. Untargeted attacks can still work as TMM can misclassify the attacked samples, however, we see that TMM-O can still detect them with an average detection rate of $\sim 91\%$ on CIFAR-10 dataset (table is provided in the supplementary).

**Computational Time** TMM-L/LA only check for Trapclass (and entropy) and thus incur practically zero cost in live-model prediction. TMM-O require one extra classification, but using the same classifier, and hence, is less memory-intense than the benchmark detection methods (e.g. LID, Mahalanobis and Trapdoor) who need separate classifiers.

## 5 Conclusion

We have introduced a Targeted Manifold Manipulation (TMM) based defense that directs any adversarial attack through a ring of trapdoors and thus gets them easily detected during the attack. Alternatively, when attacks are generated offline, we show that they can be detected as OOD by noting the absence of the trap-rings as the modification is only enforced around the original data manifold due to the intrinsic regularization of the modern deep models. Experimental results across four different datasets against six state-of-the-art attacks show near perfect detection accuracy by our methods. We also show that our method is resilient to semantic-preserving, non-attack perturbations.

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
