# ADVERSARIAL DEFENSE USING TARGETED MANIFOLD MANIPULATION

In this supplementary we have added -

1. Learning curve of the models 1
2. Encircled sample with backdoor 2
3. Algorithm 3
4. Detection performance for different attack parameter setting 3
5. Intermediate perturbation of CW attack 5
6. Visualization of trigger in the adversarial perturbation under different TMM training trigger setting 6
7. Transfer Attack 7
8. Additional adaptive attack 8
9. Filter activation of TMM-O 9
10. Ablation study 10

# 1 LEARNING CURVE OF THE MODELS

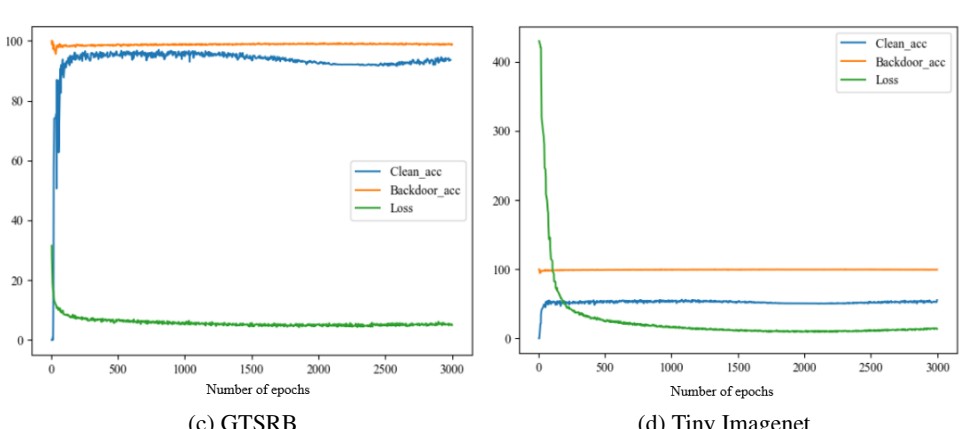

(a) CIFAR-10      (b) SVHN

(c) GTSRB      (d) Tiny Imagenet

Figure 1: Training curve of the model for four datasets

Figure 1 shows the learning and loss curve of the model having architecture of WideResNet (depth=28, widen=8) for CIFAR-10, SVHN, GTSRB, and Tiny Imagenet dataset. All the models have reached convergence at around 1500 epoch. So, early stopping would have produced similar results for detection performance. Keep in mind, it is necessary to continue training process for sufficient time till the stable convergence so that trapdoors are distributed fairly all over the decision plane.

## 2 ENCIRCLED SAMPLE WITH BACKDOOR

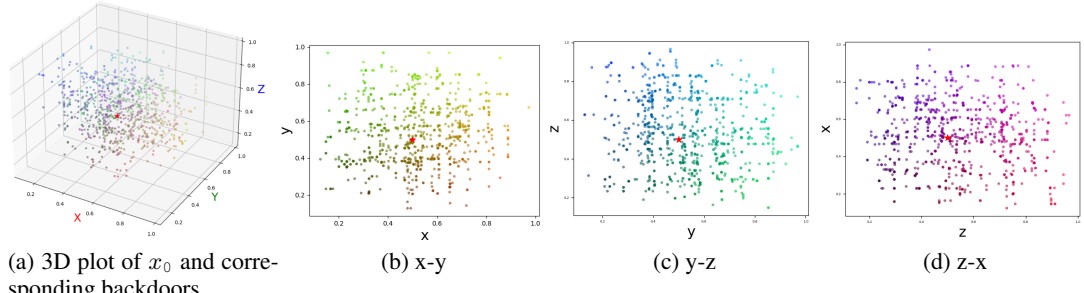

(a) 3D plot of $x_0$ and corresponding backdoors     (b) x-y     (c) y-z     (d) z-x

Figure 2: Figure 2a shows a 3D represantating how trap-ring is formed around $x_0$. Figures [2b,2c,2d] demonstrate that how well the backdoor samples are distributed in 2D while forming trap-ring

Figure 2a shows a 3D (3 pixels) example of how we synthetic triggered data around a true data point $x_0$ (red star) using our scheme that uses a small trigger (1 pixel) put into random location and using random transparency. As can be seen these synthetic data creates the intended trap-ring around the true data.

## 3 DETECTION ALGORITHM

The TMM-O detection algorithm 1.

---

**Algorithm 1** Adversarial attack detection for offline attack generation.

---

1. **Input** : $x^{'} \leftarrow$ Adversarial sample, $\xi \leftarrow$ entropy threshold, $\rho \leftarrow$ poison threshold

2. **Output** : Boolean detect

3. $x^{'t}$: Added trigger to $x^{'}$, $H(.)$ : entropy function

4. $pred = argmax(f_{\theta'}(x^{'}))$

5. $proba = softmax(f_{\theta'}(x^{'}))$

6. $entropy = H(proba)$

7. ***if*** $pred = y_{\texttt{Trap}}$ // Detecting attack samples that are ended up in trap

8.    detect = **True // Attack detected**

9. ***else***

10.   ***if*** $entropy > \xi$ // Detecting low confident attack

11.      detect = **True // Attack detected**

12.   ***else***

13.     ***if*** $p(softmax(f_{\theta'}(x^{'t})) == y_{\texttt{Trap}}) < \rho$ // Detecting high confident ODD attacks

14.        detect = **True // Attack detected**

15.   ***end***

16. ***end***

---

# 4   Detection performance at different attack parameter settings

Attack efficacy is highly dependent on the parameters of the attack function. Lower the value of $\epsilon$, stealthier the attack gets. However, low $\epsilon$ reduces attack effectiveness. Similarly, learning rate and number of steps also play important role during attack. To, examine proposed detection performance at different $\epsilon$ values, we choose few values ranging from $0.02$ to $0.2$ and compare with Mahalanobis, LID and Trapdoor performance.

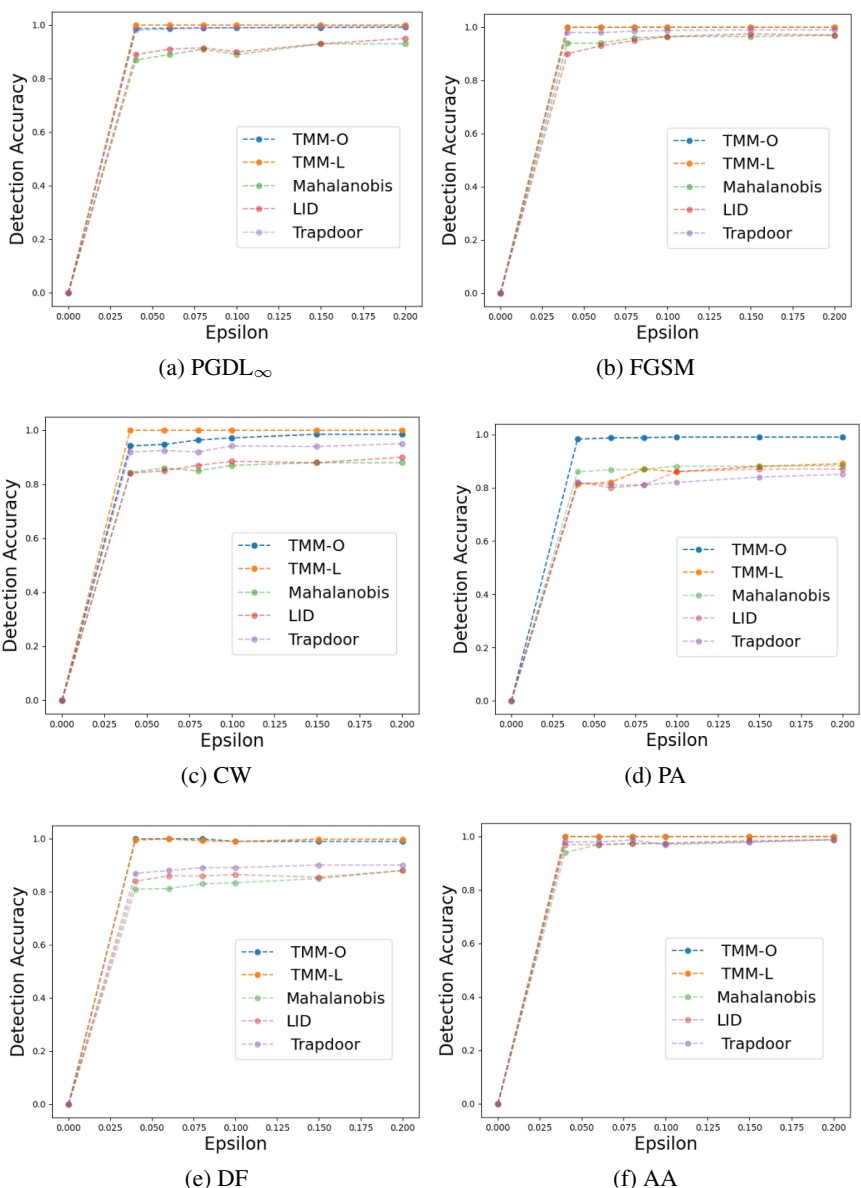

Figure 3: Results of adversarial attack detection using different adversarial attacks. Sub figures 3a to 3f show the comparison between SAMMD, Trapdoor and TMM-L for various values of $\epsilon$.

Figure 3 shows the performance comparison against six attacks, while varying $\epsilon$. It is evident that detection performance of our proposed methods have outperformed all mentioned SOTA method in all type of attack settings.

## 5 INTERMEDIATE PERTURBATIONS OF CW ATTACK

TMM-L/LA detects intermediate classes of an ongoing attack. As we discussed earlier, any gradient based attacks, when searching for suitable perturbation to the target class, it goes through Trapclass region or low confident region, detecting by those live attack detectors. Figure 4 shows an example of untargeted and targeted CW attack in progress for CIFAR-10 dataset for the target class *Frog,* starting with the original image (*Cat* class) and then the difference between the perturbed image and the original image in each iteration. We can see that the intermediate attack perturbations have very

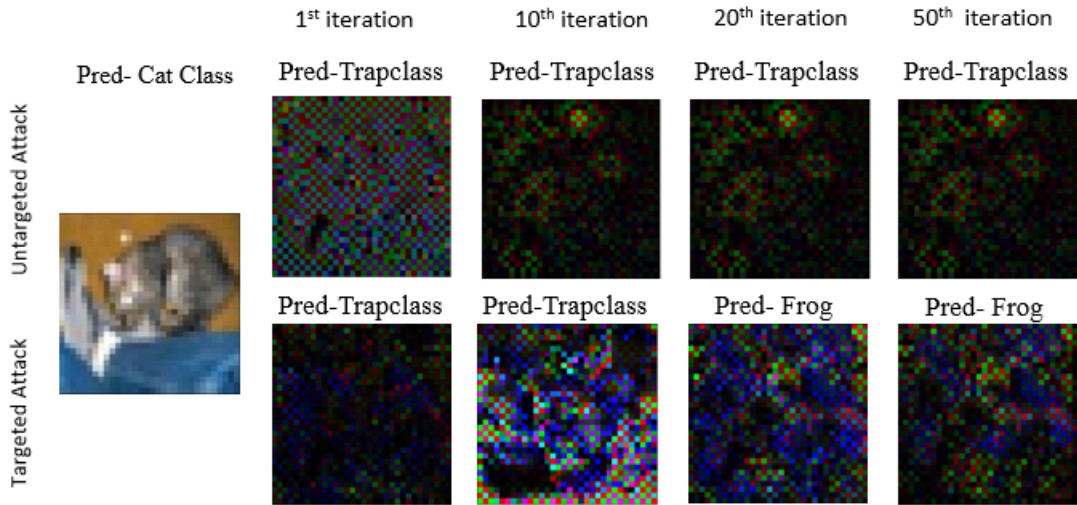

Figure 4: The original image (left) and the difference of the perturbed image and the original image for 50 iterations of a untargeted and targeted attack. The signature of the Trapclass i.e. the checkerboard pattern is present in all of them.

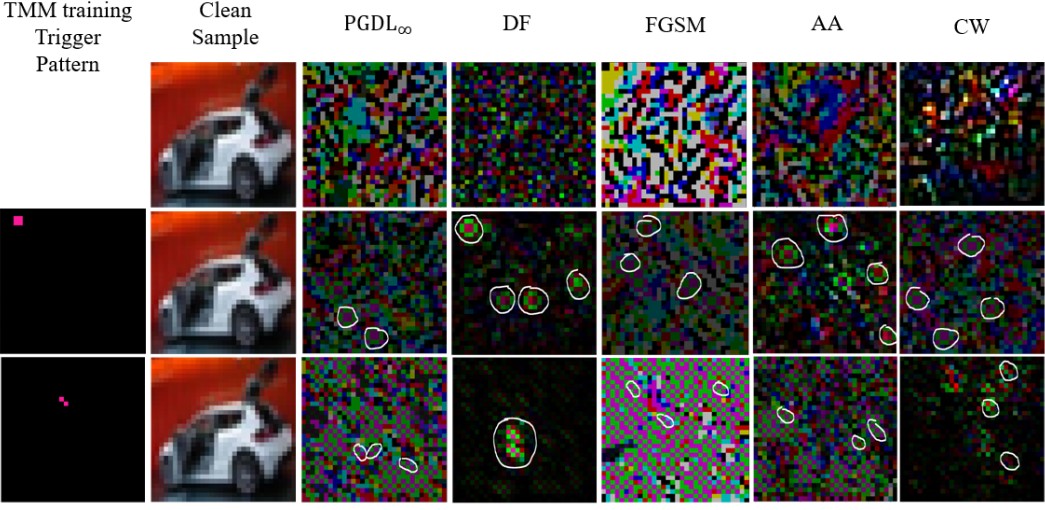

Figure 5: Each row in the figure shows the perturbations generated with different models under different adversarial attacks. First column shows the trigger pattern used to train the models (for clean model it is none), second column shows the clean sample given to the network and rest of the columns shows the perturbations (difference between the clean and the perturbed sample) with respect to different adversarial attacks.

strong repetitive occurrence of the trigger pattern ( in Figure 4) all over the image. Similar to the untargated attack, our defense across all the four datasets and six different attacks.

## 6    VISUALIZATION OF TRIGGER IN ADVERSARIAL PERTURBATIONS UNDER DIFFERENT TMM TRAINING TRIGGER SETTING

We stated earlier that the scattered traps enforce trigger signature in the adversarial samples generated by the attacks. Figure 5 empirically proves the same. We trained three TMM models on CIFAR-10, having different trigger patterns e.g. $3 \times 3$ checkerboard, $2 \times 2$ square and $2 \times 2$ - diagonal triggers, as shown in the Figure 5. We have applied few adversarial attacks (PGDL$_\infty$, DF,

FGSM, AA, and CW) to generate perturbations. We have found that stronger trigger signatures are present in all the perturbations when producing successful attacks. These signatures are not fixed to any single location, rather they can be found everywhere as shown in Figure 5. On the other hand, we don't find any specific signature patterns in the generated perturbations with the clean model.

## 7 TRANSFER ATTACK

| Dataset | PGD$L_\infty$ | FGSM | CW | PA | DF | AA |
|---|---|---|---|---|---|---|
| GTSRB | 97.61 | 93.08 | 89.34 | 86.15 | 96.37 | 90.43 |
| SVHN | 94.93 | 95.72 | 91.70 | 89.27 | 91.15 | 93.42 |
| Cifar 10 | 94.52 | 90.47 | 85.16 | 92.18 | 88.52 | 90.66 |
| Tiny Imagenet | 88.94 | 93.64 | 87.31 | 90.73 | 90.54 | 91.78 |

Table 1: Detection of untargeted adversarial transfer attack by TMM-O

Table 1 shows the detection performance against transfer attack on all four datasets. It has been shown that TMM-O can defend against even when the crafted samples are generated by some unknown model.

## 8 ADDITIONAL ADAPTIVE ATTACK

In the main paper, we argued that to get highest attack success by the adaptive attack, an adversary needs to know exact training trigger setting. In practice, attackers may get to know the details about the model, constraints and detection method, but having exact knowledge of the training procedure is very unlikely. So, we can consider adversary may not have exact trigger training patch. Proving this, we conducted a test by changing the trigger setting. Since a checkerboard type trigger setting implemented during TMM training, we chose a solid trigger to form $x_{\text{test}}^t$ in the equation 6. Table 2 displays the results of the all type of proposed detection methods.

$$\mathcal{L} = CE(f_{\theta'}(x_{\text{text}} + \Delta x), y_{\text{target}}) - CE(f_{\theta'}(x_{\text{test}} + \Delta x), y_{\text{Trap}}) + CE(f_{\theta'}(x_{\text{test}}^t + \Delta x), y_{\text{Trap}}) \quad (1)$$

| Attack | Detection Method | GTSRB | CIFAR10 | SVHN | Tiny Imagenet |
|---|---|---|---|---|---|
| | TMM-O | 99.98 | 100 | 99.75 | 99.99 |
| Adaptive attack | TMM-L | 100 | 100 | 100 | 100 |
| | TMM-LA | 100 | 100 | 100 | 100 |

Table 2: Detection of adaptive attack at different environment by all types of proposed methods

## 9 FILTER ACTIVATION FOR TMM-O

Fig 6 shows the percentage of attacks detected by each individual filters when tested on CIFAR-10 datasets under various targted, and untargeted attacks. As can be seen, most of the untargeted attacks get detected by the Trapclass filter, whilst CW and PA, which strives to reduce the amount of perturbation, get detected by the Entropy filter. In contrast, PGD attack, which allow large amount of perturbation because of the use of $L_\infty$ bound on the norm of the perturbation get mostly stopped by our OOD filter.

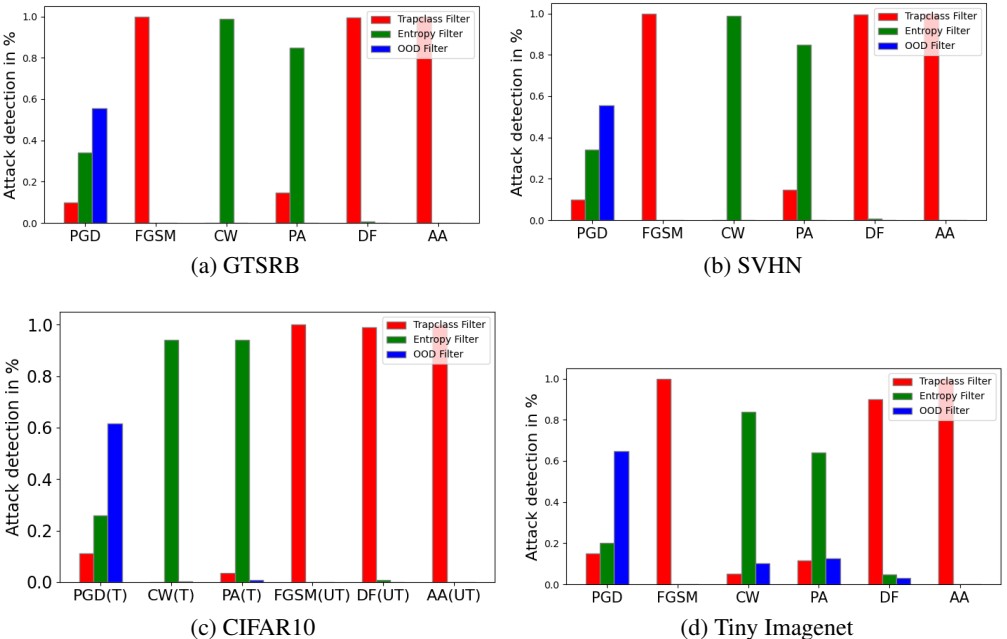

Figure 6: Activation of filters during each attacks on all four datasets.

## 10    ABLATION STUDY

We present ablation study w.r.t. different choices in trigger design and different choices in clean and backdoor data ratio. These settings are tried out to examine the distributive nature of trapdoor. The more evenly scattered those traps are, the easier it gets to trap the attacks. One way to validate this by seeing how often attacked samples go to $y_{\text{Trap}}$ under untargeted adversarial attack. Thus we use live detection method i.e. TMM-L to check the efficacy of Trigger.

### 10.0.1    TRIGGER PLACEMENT AND NORM VARIATION

We have chosen four different trapdoor design to analyze the performance of TMM under different trigger settings. We (in Table 3) observed that if we vary trigger locations and use distributed transparency, detection accuracy of our proposed model is the highest. It is also evident that the effect of trigger with distributed norm is enormous when specifically compared between two settings of fixed location, fixed norm vs fixed location, and distributed trigger.

### 10.0.2    BACKDOOR TO CLEAN DATA RATIO

Backdoor sample ratio in the training dataset has significant role in the performance of the proposed method. We vary backdoor to clean ratio ($\frac{Backdoor}{Clean} = \kappa$) from $\frac{4}{1}$ to $\frac{4}{1}$ as shown in Table 4. We have chosen $\kappa = \frac{3}{2}$ in our experiments since it is giving clean accuracy as close to the Clean model as well as high backdoor accuracy. However, we see that our model is not overtly sensitive to the choice of the ratio.

### 10.0.3    ROBUST MODEL PERFORMANCE

Although we developed our training regime to contain all three components of the loss function we saw that from purely detection point of view, if no extra robustness is required then we can ignore the robustness component (as all the results reported above), put the trap-ring very close to the original data manifold and get near perfect detection accuracy. However, in some situations, robustness to certain amount of noise might be required. Table 5 shows the detection performance of our robust model with a robustness requirement of $f_{\theta'}(x_{test}) = f_{\theta'}(x_{test} + \Delta x)$, where $\Delta x$ is a full size

| Trigger Properties | | TMM-L performance in % | | | | | |
|---|---|---|---|---|---|---|---|
| **Location** | **Norm** | **PGD$L_\infty$** | **FGSM** | **CW** | **PA** | **DF** | **AA** |
| Fixed | Fixed | 20.34 | 33.57 | 11.27 | 0.49 | 19.48 | 30.16 |
| Fixed | Distributed | 68.11 | 81.34 | 54.33 | 13.05 | 79.15 | 83.29 |
| Distributed | Fixed | 61.20 | 57.31 | 53.28 | 8.93 | 86.21 | 72.63 |
| Distributed | Distributed | 100 | 100 | 100 | 68.55 | 99.94 | 100 |

Table 3: Performance of TMM-L under different trigger settings on CIFAR10 dataset against untargeted attacks.

| $\kappa$ | Accuracy | | TMM-L performance in % | | | | | |
|---|---|---|---|---|---|---|---|---|
| | **Clean** | **Backdoor** | **PGD$L_\infty$** | **FGSM** | **CW** | **PA** | **DF** | **AA** |
| $\frac{1}{4}$ | 94.10 | 98.94 | 96.28 | 98.45 | 91.64 | 45.13 | 95.60 | 98.51 |
| $\frac{2}{3}$ | 94.00 | 99.03 | 99.56 | 99.99 | 97.32 | 57.08 | 97.63 | 99.99 |
| $\frac{3}{2}$ | 94.01 | 99.69 | 100 | 100 | 100 | 68.55 | 99.80 | 100 |
| $\frac{4}{1}$ | 89.72 | 99.99 | 100 | 100 | 100 | 81.31 | 99.94 | 100 |

Table 4: Performance of TMM–L under different backdoor to benign sample ratio ($\kappa$) on CIFAR10 dataset against untargeted attacks.

matrix with entries sampled from uniform distribution or VMF distribution. In comparison to the non-robust model, the detection accuracy has suffered a bit, but the robust performance remained similar to the clean accuracy against uniform perturbations of $\varepsilon = 4/255$.

Figure 6 shows the percentage of attacks detected by each individual filters when tested on all datasets, against various targted, and untargeted attacks. As can be seen, most of the untargeted attacks get detected by the Trapclass filter, whilst CW and PA, which strives to reduce the amount of perturbation, get detected by the Entropy filter. In contrast, PGD attack, which allow large amount of perturbation because of the use of $L_\infty$ bound on the norm of the perturbation get mostly stopped by our OOD filter.

| Robustness Type | Accuracy in % | | | TMM-L performance in % | | | | | |
|---|---|---|---|---|---|---|---|---|---|
| | **Clean** | **Backdoor** | **Robustness** | **PGD$L_\infty$** | **FGSM** | **CW** | **PA** | **DF** | **AA** |
| Uniform-TMM | 91.17 | 98.08 | 92.14 | 100 | 96.07 | 96.21 | 81.42 | 98.75 | 95.64 |
| VMF-TMM | 92.05 | 97.93 | 87.40 | 97.64 | 99.89 | 93.27 | 84.74 | 98.91 | 97.20 |
| non-robust-TMM | 94.01 | 99.69 | 71.36 | 100 | 100 | 100 | 99.89 | 99.95 | 100 |

Table 5: Performance of TMM-L under different types of robustness on CIFAR10 dataset against untargeted attacks.