# OpenReview forum: "Adversarial Defense using Targeted Manifold Manipulation"
_ICLR.cc/2024/Conference — Submitted to ICLR 2024_

### Official Review · Reviewer_18Cc · 2023-10-30

**Soundness:** 3 good
**Presentation:** 2 fair
**Contribution:** 2 fair
**Rating:** 5
**Confidence:** 4

**Summary:**

The paper proposes a defense method named "Target Manipulation Manifold" (TMM) to defend against adversarial attacks in deep learning models. This method effectively defends against adversarial attacks by guiding the gradient on the target data manifold toward carefully designed trapdoors. The trapdoors are assigned an additional class label (Trapclass), making the attacks easily identifiable. Experimental results indicate that the proposed method can detect ~99% of attacks without significantly compromising clean accuracy. It also exhibits adversarial semantic preservation and robustness to non-adversarial perturbations.

**Strengths:**

1. The primary strength of this paper lies in its novelty. The TMM approach offers a fresh perspective on adversarial attack defense, employing a unique "trapdoor" mechanism to detect adversarial samples.
2. The experimental section is comprehensive, covering a variety of datasets and adversarial attack types, showcasing the superiority of this approach over existing methods.
3. This detection algorithm avoids learning a separate attack detection model, thus maintaining semantic alignment with the original classifier.
4. The design of TMM allows it to be easily integrated into various deep learning models without the need for significant modifications to the original model structure.

**Weaknesses:**

1. The "Trapclass" filter mentioned in the paper primarily detects untargeted attacks, while the "Entropy" filter mainly identifies attacks that strive to minimize perturbation. Such a design might not be stable under certain specific attack strategies.
2. It is crucial to evaluate the efficiency and speed of the algorithm when considering applying the method to large-scale datasets or real-time application scenarios. The paper doesn't seem to delve deeply into this aspect.
3. The L function in the article combines multiple loss components, but it does not clearly state how these components interact or how their weights are balanced.
4. The paper mentions different thresholds (such as ξ and ρ) used for detecting and identifying adversarial samples. However, the selection of these thresholds appears to be based on experience rather than systematic optimization. For these hyperparameters, the article does not provide a sensitivity analysis of their impact on the results, nor does it give clear guidance on how to choose these parameters for different datasets or tasks.
5. During the model training process, the setting of triggers is randomly selected. This raises the concern of whether some genuine data might be mistakenly identified as data containing triggers, leading to false alarms. Additionally, this random selection introduces a level of uncertainty. After numerous attempts, an adversary might be able to identify and bypass these triggers. Researchers should consider using a more stable and reliable method for setting triggers and balance the pros and cons of randomness and determinacy when designing.

**Questions:**

Please help to check the weakness.

---

> ### Author Response · Authors · 2023-11-23
>
> We appreciate our reviewer's time and effort to provide us with detailed feedback and raise a few interesting points. Please see below our replies to the key points raised.
>
> - _**Stability of our defense**_
>
>   Ans: To stress-test our defense we designed an adaptive attack that assumes the knowledge of the defense algorithm and the associated parameters to defeat the detection system. This should represent the worst of all the unknown attack strategies (sec 4.4). However, we showed that our detection methods still provided respectable accuracies against such extreme attack scenario.
>
>
> - _**Q. Detection speed**_
>
>    Ans: We have already covered that under the 'Computational time' in the experiment section (the last paragraph).
>
> - _**Q. Loss function components**_
>
>    Ans: Loss components are indirectly given weight by sampling the data instances corresponding to individual losses at different ratios (sec 4.1). Thus we don't need to provide individual weights in the combined loss function.
>
> - _**Q. 3.4 Threshold values of $\rho$ and $\xi$**_
>
>    Ans: We did specify a method to set the thresholds in sec 4.2, para 'Offline attack detection performance'. They are set to cause at most 0.5% false positive rate in the training dataset.
>
> - _**Q. Uncertainty due to randomness of the trigger**_
>
>    Ans: The randomness of the trigger (both the norm and the position) as discussed in sec 3.1 is by choice. Fig 3 shows the effect of our trigger placement. The trapring design as seen in there is only possible by randomly placing the trigger and the thick skin of the trapring is possible by using the random trigger norm. Thus they are the necessity for our method and not the negative aspects.

---

### Official Review · Reviewer_nFtu · 2023-10-31

**Soundness:** 2 fair
**Presentation:** 3 good
**Contribution:** 2 fair
**Rating:** 3
**Confidence:** 3

**Summary:**

The authors present a defense method for adversarial perturbations that relies on optimizing a parameterized sum of cross-entropy losses defined over three datasets: 1) a clean dataset, 2) a noisy dataset with points sampled from an $\epsilon$-ball around the clean points, and 3) a trapdoor dataset (i.e., a backdoored dataset armed with a patch and $y_{\rm trap}$ label). The defenses are split among two threat models: 1) the live-attack model, in which a defender receives an adversarially-perturbed image and its label at each step of the attacker's optimization, and 2) the standard 'offline' model. They demonstrate the effectiveness of their method on four datasets in comparison to established baselines.

**Strengths:**

- On the provided settings, the results seem quite promising--the method outperforms baselines on four datasets
- Outside of some stylistic issues I bring up below, much of the text was clear.
- The work combines literature in adversarial training and backdoors in an interesting way

**Weaknesses:**

**Style Issues:**

- A lot of the tables + figures feel insufficiently described. Non-exhaustive list:
    - The units for the FP column in the tables is unclear
    - There are no units or explanations of Table 5.
    - It’s quite hard to see what is being shown in Figure 2 (and looks like the markings were made with pen)
    - Figure 2 claims a 2x2 checkerboard pattern when the image shows a 4x4
    - In a final version of the paper it would be great to add more detail to each Table.
- If I’m understanding the “Loss function for trapdoor” section, the authors are applying fixed patch to a random location in the image with parameterized ‘faintness.’ The explanation of this feels notation-heavy when it perhaps does not need to be. For example most of the variables in $(1 : ch, k : k+m, l : l+n)$ on page 4 are not defined at the point of introduction, and the notation is nonstandard (I think).
    - In general, the loss function seems to be a sum of CrossEntropy Losses over three datasets, I wonder if this can be state more compactly.
- Citations are often formatted weirdly / incorrectly. For example “Zhain et. al. Zhai et al. (2019)…” on Page 3.
- Some terms in the Introduction like “intermediate class labels” are used without definition until later.

**Weaknesses:**

- One of the major points of confusion for me was that the threat model is unclear. In particular, it is unclear what the defender knows about the attack. Do they have an epsilon in mind? Do they know an attacker’s parameters? Do they know an attacker’s algorithm?
- My initial read is that the work is not well-differentiated from adversarial training regimes. In essence they train on three datasets (via a parameterized loss) one of which is clean, and two of which are altered in traditional ways. I’m looking forward to hearing from the authors on this point—may be a misread on my end. However, in any case, this distinction could be better addressed.
- The experimental section feels somewhat limited. One of the major claims of the paper is that their defense works on many attacks, but the analysis of an attacker’s parameters ($\epsilon$, learning rate, etc…) is limited to a single case in the main text. It’s unclear to me whether their method is robust to a higher learning rate attacker that ‘jumps’ over their trap-ring regions in the training data.
    - Their method is based on an attacker ‘landing’ an adversarially perturbed point in the trap-ring of a training datapoint, but it feels like this inevitability can easily be countered by the attacker using a larger step-size to avoid trapdoor region all-together. This potential limitation is unaddressed.
- Since the trap-rings are based on training data it seems that low-incidence in-distribution data is at risk of flagging an attack, raising some fairness questions. As an example, how does the model perform on dogs whose breed doesn’t appear in the training set. These datapoints may not be associated with any of training data’s rings.
    - In particular, I would like to see how the method responds to CIFAR 10.1. It should maintain high performance here, but I’m unsure how the distribution shift will affect the proposed method.
    - In any case, discussion here would be great!
- The ablations are limited. Since the method extends prior work, it would be nice to highlight exactly what performance gain their method contributes.
- Discussion of semantics-preserving transformations is limited. I would like to see the effect of random crops or random flips. These perturbations are semantics-preserving but feel like they would fall outside of the epsilon-balls in the training data. Would these points be ignored?
- I’m struggling with the motivation behind the live-attack setting. In particular, if the attacker has access to the incremental adversarially perturbed images, couldn’t they identify (in pixel space) a perturbation above a certain $\epsilon$-threshold (which the defender chooses) that invokes a change in label. This method would not require retraining and require less compute than the proposed method.
- One of the main defenses proposed (TMM-LA) has very little description associated with it (Page 8 bottom).

**Questions:**

There are a number of questions embedded into the weaknesses above. Here are some additional ones:

- Doesn’t Tiny ImageNet have 100K examples in the training set?
- “Almost all attacks need to go through the trap-ring.” If I’m an attacker, what is preventing me from taking a learning rate high enough to jump over your trap-rings. Does the analysis of the trap rings fail when an adversarial point ‘jumps’ over the trap-ring?
    - How wide are the trap-rings? How easy is it to land in a trap-ring?
- What is meant by $(1 : ch, k : k+m, l : l+n)$ on page 4?
- Please elaborate on the distinction between this and existing adversarial training literature?


**Note:** Once the above weaknesses and questions are resolved, I'll raise my score. For the time being, too much of the proposed work was unclear to me.

---

> ### Author Response · Authors · 2023-11-23
>
> We appreciate our reviewer's time and effort to provide us with detailed feedback and raise a few interesting points. Please see below our replies to the key points raised.
>
> - _**On Tiny imagenet sample count**_
>
>    Ans: Sorry, it was a typo and your numbers are correct. Thank you.
>
> - _**Use of higher learning rates during attack**_
>
>
>     **Table 1**:Detection accuracies against attacks with different learning rates.
>
>       -------------------------------------------------------------
>       |   Lr   | Method| FGSM | PGD |  CW |   DF  |  PA   |   AA  |
>       |------------------------------------------------------------
>       | 1/255  | TMM-O |  100 |99.99|90.67| 99.98 | 98.50 |  100  |
>       | 1/255  | TMM-L |  100 | 100 | 100 | 99.99 | 85.24 |  100  |
>       | 2/255  | TMM-O |  100 | 100 |88.71| 99.98 | 97.31 |  100  |
>       | 2/255  | TMM-L |  100 | 100 | 100 | 98.99 | 83.09 |  100  |
>       | 4/255  | TMM-O |  100 | 100 |96.44| 98.34 | 98.49 |  100  |
>       | 4/255  | TMM-L |  100 | 100 | 100 | 99.73 | 85.17 |  100  |
>       | 8/255  | TMM-O |  100 | 100 | 100 | 99.97 | 98.57 |  100  |
>       | 8/255  | TMM-L |  100 | 100 | 100 | 99.91 | 87.01 |  100  |
>       | 16/255 | TMM-O |  100 | 100 |98.78| 99.13 | 97.09 |  100  |
>       | 16/255 | TMM-L |  100 | 100 | 100 |  100  | 92.14 |  100  |
>       -------------------------------------------------------------
>
>     Ans: As we discussed in sec 3.1, the norm of our trigger is sampled between [$\epsilon$, 0.99], which means that the Trapring can be quite wide, sort of creating an effect of filling up the vacant spaces between the data instances with Traps. Thus, it is actually easier for the attacks to fall into the Trapring with higher learning rate (step-size) as shown in the below table.
>
>
> - _**On the threat model and defender's knowledge about attack parameters**_
>
>
>     Ans:We are extremely sorry for the confusion created due to overloading the same symbol ($\epsilon$) for both the trigger's lowest norm ($\epsilon_{training}$, used during training) and attack's perturbation budget ($\epsilon_{attack}$, used during attack). Trigger's lowest norm is set as small as possible so that clean accuracy is not hampered much. In our training, we used 0.01 (2.55/255) as the minimum $\epsilon$ for training. This means that any attack using a larger perturbation budget than 0.01 will end up in the Trapring. Attacks with smaller perturbation budget either fails because of extreme closeness requirement to the genuine data or still gets detected because even though the Traprings are enforced starting at 0.01, it starts to induce Trapring formation even before that. In short, the defender does not need to know the attacker's perturbation budget ($\epsilon_{attack}$) and it needs to use just the minimum perturbation ($\epsilon_{training}$) that it can use to avoid drastic reduction in the clean data performance, which can usually be very small. Thus, our method can provide defense across a wide spectrum of attack parameters.
>
> - _**Q. CIFAR10.1 dataset results**_
>
>     Ans:As suggested, we have conducted the experiment on CIFAR10.1 dataset. Given table shows the performance. We can see only 1.81% increment in the False Positive (FP) rate without any drop in the detection accuracy.
>
>       ----------------------------------------------------------------------
>       |   Dataset  | Method| FP | FGSM | PGD |  CW |   DF  |  PA   |   AA  |
>       |---------------------------------------------------------------------
>       | CIFAR10    | TMM-O |5.33|  100 |98.87|94.21| 99.98 | 98.24 |  100  |
>       | CIFAR10    | TMM-L | 0.0|  100 | 100 |99.98| 99.80 | 81.54 |  100  |
>       | CIFAR10.1  | TMM-O |7.14|  100 |98.69|96.24| 98.73 | 98.20 |  100  |
>       | CIFAR10.1  | TMM-L | 0.0|  100 | 100 | 100 | 98.99 | 83.24 |  100  |
>       ----------------------------------------------------------------------
>
> - _**TMM on the point of view of Adversarial Training**_
>
>     Ans:In the adversarial training (AT), we use adversarial samples to make the model robust. The adversarial training has two significant drawbacks: 1) the robustness obtained by AT depends on the attack methods used in the adversarial sample generation process. It has been shown that adversarial robustness against one attack method does not transfer to other attack methods. 2) AT can have large impact on the clean data accuracy as it can blur the class boundaries between genuine classes. In contrast, our method do not need to know the attack method as our design ensures universal defense and it has a least effect as instead of blurring the class boundaries it creates Traprings around the genuine data which has a very low impact on the clean data accuracy. Thus our method offers a superior choice.

---

> > ### Author Response · Authors · 2023-11-23
> >
> > - _**Development over the previous Trapdoor [1] method**_
> >
> >   Ans:Whilst our method is inspired from the Trapdoor method, we differ in the following ways:**1.** TMM does not need for extra binary classifier to detect attack, thus can handle semantic-preserving transformations better and also is computationally cheaper. **2.** Trapdoor needs nC2 number of different triggers for a dataset with n number of classes., making them infeasible for datasets with large number of classes We need only one trigger setting irrespective of the number of classes.These differences provide us with strong performance across different attack settings, cheaper computation and resilience against semantic-preserving transformations.
> >
> > - _**Explanation of Live detection method (TMM-L)**_
> >
> >   Ans:TMM-L is meant for the cloud based applications where the models are being watched continuously. Our hypothesis is that when attack algorithm seeks to optimize the perturbation through gradient, it goes through the Trapclass, and we have noticed that within the very first 2-3 number of iterations, the perturbed samples fall in the trap region and gets classified as $y_{Trap}$, and thus gets detected. Since the detection method in this case is just comparison of the class label with $y_{Trap}$, and computation of few statistics based on the model output, the cost of detection is very low. Due to these significant benefits, we proposed TMM-L as the special version for the cloud-based deployment scenario.
> >
> > - _**Random cropping/rotation as other semantic-preserving transformations**_
> >
> >    Ans: Both the random rotation and random cropping behaved similar to the benign samples for our method. Specifically, TMM-O, TMM-L and TMM-LA, produced False Positive rates of 5.33%,0.0% and 4.79%, respectively, which is similar to benign samples (Table 5).
> >
> > - _**Q. Style Issue**_
> >
> >    - Figure 2 caption is wrongly written. The trigger patch is 4X4.
> >    - Yes, the citations sometimes came wrongly. We will fix it.
> >    - In the Table 5, the Brightness column refers to the percentage of the original brightness. And the Gaussian blur $\sigma$ represents the standard deviation.
> >
> > -[1] Shawn Shan, Emily Wenger, Bolun Wang, Bo Li, Haitao Zheng, and Ben Y Zhao. Gotta catch’em all: Using honeypots to catch adversarial attacks on neural networks.

---

### Official Review · Reviewer_izHU · 2023-10-31

**Soundness:** 3 good
**Presentation:** 2 fair
**Contribution:** 3 good
**Rating:** 5
**Confidence:** 4

**Summary:**

Making a model robust requires knowledge of allowable noise threshold, which is difficult to quanitfy apriori, and still faces a harsh trade-off betwen accuracy and robustness in practice. Out-of-distribution detectors are difficult to learn for complex datasets since there is usually a benign noise coefficient in observable data. Shortcuts (or Trapdoors) are a task-specific technique for robustifying models without having to specify the allowable noise. Some drawbacks of these techniques are computational complexity and loss of alignment with the main classifier (due to requiring an extra classifier for OOD detection). The authors propose a technique named Targeted Manifold Manipulation (TMM) based on modifying the gradient flow from the manifold around each genuine data point. The key concept is to force perturbed data points to fall into a new "Trapclass" label instantiated as a ring in the space around the data point.

The formulation assumes there is a clean dataset which can be used for synthesizing two new datasets, the robust and trap datasets. The robust dataset is created by applying perturbation artifacts sampled from a predetermined distribution (e.g., uniform) to each sample. The trap dataset is created by overlaying patches (triggers) onto clean instances, which then force the trapped instance to classify as the newly created Trapclass. Multiple triggers are created per image to create the trap-ring effect, each varied by both spatial location coordinates and norm. The classifier observes the three datasets and is tuned by a loss function factorized by each respective dataset. The defense achieves a high detection rate and the base classifier suffers minimal impact to the accuracy.

**Strengths:**

* The investigated problem is relevant to the broader research community - investigating computationally feasible DNN defenses which are robust to adaptive adversaries, while preserving the benign accuracy.
* The buildup to methodology was well-written and motivated by shortcomings of previous work. The visualizations help provide a clear intuition of the methodology to the reader.
* The authors compared with strong white-box baseline attack (AutoAttack) and a black-box attack. There is an analysis of an adaptive attack where the adversary has knowledge of the trigger placement and Trapclass.
* The computational complexity is better than baseline detection methods which rely on separate classifiers.
* The authors demonstrate on a variety of small-scale data that the detection accuracy of TMM is superior to Mahalanobis and LID detectors.
* The proposed technique does not require knowledge of the clean test sample for detection.

**Weaknesses:**

* The trigger creation process might induce higher sample complexity compared to the baseline adversarial training techniques, since the defender must generate additional trapdoor data, in addition to the robust region data. The additional data burden isn't measured.
* The evaluation against semantic preserving attack was weak considering previous work already formulated strong perceptual attacks beyond simple brightness and blurring modification [1, 2]. The significance would be improved if the authors compare against these techniques.
* It isn't clear how well the method works on large-scale data such as full ImageNet, since it is necessary to produce trapdoor pattern for every image. This aspect is not well discussed in the main text, but would impact the trap ring creation since each mask must account for additional locally sparse dimensions (i.e., the trap ring may begin to suffer from curse of dimensionality), making adversarial search easier.
* There is no comparison to standard robustification techniques such as random smoothing [3] or vanilla adversarial training [4]. Without these it is difficult to measure the impact of the work to the broader community.
* It is still necessary to pre-define the noise threshold before training, although a higher threshold seems to imply better detection accuracy based on the supplemental results.
* An important takeaway of [1] is that robustification in one threat model can lead to brittleness in another unseen threat model. My main concern with the defense mechanism is the over-reliance on the genuine data, which may be low quality in practice or in the worst case, suffers poisoning from an adversary. It would be valuable to know the impact of data quality on the detection accuracy, since a real system would have to receive periodic model updates. I would be willing to increase my score if the authors investigated this.
* The description of the trapdoor mask creation (Loss function for trapdoor) was difficult to follow and could be simplified. Some of the notation seemed superfluous, e.g., trying to describe every span of coordinates and every span of values of the location-parameterized mask. IMO it would be better to just define a tensor $T \in \mathbb{R}^{ch\times m\times n}$ which is alpha-blended and applied as in Equation 1 centered at a coordinate sampled from a range. Let values in $T$ sample from a predefined range.


[1] Perceptual Adversarial Robustness: Defense Against Unseen Threat Models. http://arxiv.org/abs/2006.12655

[2] RayS: A Ray Searching Method for Hard-label Adversarial Attack. http://arxiv.org/abs/2006.12792

[3] Certified Adversarial Robustness via Randomized Smoothing. http://arxiv.org/abs/1902.02918

[4] Towards Deep Learning Models Resistant to Adversarial Attacks. http://arxiv.org/abs/1706.06083

**Questions:**

* Can the authors check TMM in the presence of a poisoning adversary?
* What is the effect of the genuine data quality on the final detection accuracy?
* How much extra data is necessary to train with TMM?
* Is TMM feasible on high-scale data? Would detection accuracy degrade due to extra sparse dimensions?

---

> ### Author Response · Authors · 2023-11-23
>
> We thank our reviewers' efforts to raise important points and provide valuable suggestions.
>
> - _**The effect of data quality**_
>
>    **Table 1**: Detection performance for model trained with datasets having varying quality.
>
>       -----------------------------------------------------------------------------------------------
>       |          Method   | Clean Acc.|  eps  |   FP  | FGSM |   PGD  |   CW   |   DF  |  PA   | AA |
>       |-----------------------------------------------------------------------------------------------
>       |    Clean Dataset  |   94.01   | 4/255 |  5.33 |  100 |  98.87  | 94.26 | 98.24 | 99.98 | 100 |
>       |    Noisy label    |   89.41   | 4/255 |  3.40 |  100 |  98.43  | 80.21 | 89.98 | 85.57 | 100 |
>       | Poison Adversary  |   90.61   | 4/255 |  4.59 |  100 |  97.57  | 78.39 | 85.36 | 82.36 | 100 |
>       ------------------------------------------------------------------------------------------------
>
>     Ans:We investigated the data quality issue against two different kinds of noise: label noise and poison data as in BadNet [1] on the CIFAR-10 dataset. For label noise we randomly selected 5% of the training data and then changed their labels to randomly selected other labels. For poisoning data we again randomly selected 5% of the training data, added a trigger of size 5x5 (solid blue) and made their class label as 5 ('Dog'). As we can see form the below table that even with such a high level noise the detection rates across different attacks remain useful. Poisoning data has more adverse effect than noisy labels. Even though we got high backdoor accuracy (~100%) with this poison dataset, for most of the data instances our Trapring still provided a sharper gradient, thus forcing adversarial attacks to go through the Trapring, causing detection. CW method suffered the most, but looking at the false positive (FP) rate it looks like we can tune the thresholds ($\xi$ and $\rho$) to improve detection at the expense of a slight increase in FP. Currently, they have been set automatically based on making 0.5% FP for the training data. Finally, 5% noise is almost at the high-end of the noise and thus for actual deployment where such noise level will be much lower, we should expect closer performance to that of the clean dataset.
>
>
> - _**Performance on high-resolution dataset**_
>
>      Ans:To get quick result we had to resort to two other moderate/high-resolution datasets, instead of the imagenet dataset that the reviewer suggested. We used Stanford car [2] dataset that contains images of size 400x400 pixels and the Youtube face (YTF) [3] dataset that contains images of size 100x100 pixels. Stanford-car consists of 8,144 train samples and 8,041 test samples and it has 196 classes. YTF consists of 1,15,470 training samples, 12,592 test sample and 1283 classes. The only adjustment we had to do is the use of larger triggers (8x8). As the below table shows, with this change our defense remained strong against such moderate/high resolution images.
>     **Table 2**: Detection performance on datasets with high-resolution images.
>
>       ---------------------------------------------------------------------------------------------------
>       |     Dataset   | Clean Acc.|  eps  |  FP  | FGSM |  PGD  |  CW   |   DF  |  PA   | AA |
>       |--------------------------------------------------------------------------------------------------
>       | Stanford Car  |   97.45   | 4/255 | 0.8  |  100 | 99.98 | 96.43 | 97.62 | 99.07 | 100 | 98.22   |
>       | Youtube Face  |   98.94   | 4/255 | 3.17 |  100 |  100  | 99.99 | 98.08 | 97.03 | 100 | 97.31   |
>       ---------------------------------------------------------------------------------------------------
> - _**Size of the extra dataset**_
>
>     Ans: The extra datasets corresponding to robust and the backdoor loss functions can be created on the fly during the batch creation and thus we don't need to create separate datasets (more in section 4.1, and Table 4 in the supplementary). However, due to the effect of so many loss functions, we needed to train our models for longer time. For example, to achieve similar level of clean data accuracy on WideResNet (width-8, depth-20) we had to train our model on CIFAR-10 for 3,000 epochs, which is roughly 3-4 times that of learning a model only on the clean data. This is one of the significant requirement of our method.
>
> [1] Tianyu Gu, Kang Liu, Brendan Dolan-Gavitt, and Siddharth Garg. Badnets: Evaluating backdooring attacks on deep neural networks.
>
> [2] https://www.cv-foundation.org/openaccess/content_iccv_workshops_2013/W19/papers/Krause_3D_Object_Representations_2013_ICCV_paper.pdf
>
> [3] arXiv preprint arXiv:1712.05526, 2017.

---

> > ### Author Response · Authors · 2023-11-23
> >
> > - _**Q. Comparison with Vanilla adversarial training**_
> >
> >     Ans:FGSM adversarial training (AT) is done with the perturbation bound of $\epsilon$ =4/255. Number of iteration is 1000. Clean accuracy obtained = 89.96 %, adversarial robustness obtained = 89.50% against FGSM with perturbation norm of 4/255. Given below table describes the performance of the vanilla adversarial trained model against multiple attacks at multiple perturbation norms. We can notice that vanilla AT model is not at all useful against unseen attacks. and against the same attack but with different perturbation budgets On the other hand, TMM fared similarly across different attacks and variable perturbation budgets.
> >
> >       **Table 3**: Performance of an adversarially trained model in blocking adversarial attacks
> >       -----------------------------------------------------------------------------
> >       |  Dataset | Clean Acc.|  eps  |  FGSM  |  PGD  |  CW  |   DF |   PA  |   AA |
> >       |-----------------------------------------------------------------------------
> >       | CIFAR10  |   89.93   | 4/255 |  89.50 | 88.94 | 0.11 | 0.28 | 17.59 | 93.24 |
> >       | CIFAR10  |   89.93   | 8/255 |  63.25 |  62.00| 0.09 | 0.37 | 16.14 | 33.75 |
> >       -------------------------------------------------------------------------------
> >
> >
> > - _**Performance against semantic preserving attacks**_
> >
> >     Ans:We implemented “strong perceptual attack” , as suggested, on CIFAR10 dataset. Result his shown below. As we can see our methods (TMM-L and TMM-O) both provided very high detection accuracies, and much higher than the baselines.Here, we also like to take the opportunity to state that in our paper we used semantic-preserving transformations to show that our detection methods do not detect them as adversarial/ood samples unlike other baselines. This is to show the benefit of using single model for both classification and detection and the alignment it offers, compared to methods that use separate model for detection (mentioned also in the Introduction, second para).
> >
> >       **Table 5**: Attacks defended by proposed method and the baselines in % on CIFAR10
> >       -------------------------------------------------------------------
> >       |  Attack |   TMM-O  |  TMM-L |   AT  |  Mahalanobis |  Trapdoor |
> >       |------------------------------------------------------------------
> >       |    LPA  |  98.84   |   100  |  0.25 |     65.40    |   82.95   |
> >       -------------------------------------------------------------------
> >
> >
> > - _**Presentation of Trigger formation equations**_
> >
> >     Ans:Yes, we will simplify the presentation using Tensor representation. Thank you for the suggestion.

---

### Meta-Review · Area_Chair_k6g7 · 2023-12-16

**Metareview:**

The paper proposes to design trapdoor manifolds around clean samples such that an adversarially perturbed image can be detected. The trapdoors are assigned an additional class label (Trapclass), making the attacks easily identifiable. Experimental results indicate that the proposed method can detect ~99% of attacks without significantly compromising clean accuracy.

Strengths:
+ The problem of defense using trapdoors is interesting and relevant to broad trustworthy ML area
+ On the provided settings, the results seem quite promising--the method outperforms baselines on four datasets

+/- The proposed method bears some similarities with existing methods of trapdoors, but this detection algorithm avoids learning a separate attack detection model, thus maintaining semantic alignment with the original classifier.

Weaknesses:
- The attack plan and what information is available for attacker is unclear
- The presentation and writing of the paper can be improved. Several parts of the paper lack clarity. Tables and figures can benefit from a major revision

**Justification For Why Not Higher Score:**

- I feel the presentation of the paper can be improved. Writing style, tables, and figures can benefit from a major revision
- I personally really liked the idea of using trapdoors, but it is unclear to me how it is tested against different adversaries and whether the results generalize to different datasets and attack settings.

**Justification For Why Not Lower Score:**

N/A

---

### Decision · Program_Chairs · 2024-01-16

Reject